# Systemic Treatment of Recurrent Hepatocellular Carcinoma after Liver Transplantation: A Multicenter Trial

**DOI:** 10.3390/cancers16132442

**Published:** 2024-07-03

**Authors:** Vera Himmelsbach, Matthias Jeschke, Christian M. Lange, Bernhard Scheiner, Matthias Pinter, Friedrich Sinner, Marino Venerito, Alexander Queck, Jörg Trojan, Oliver Waidmann, Fabian Finkelmeier

**Affiliations:** 1Department of Gastroenterology and Hepatology, University Hospital Frankfurt, 60590 Frankfurt, Germany; 2Department of Gastroenterology, Hepatology and Transplant Medicine, University Hospital Essen, 45147 Essen, Germany; 3Department of Medicine II, University Hospital, Ludwig-Maximilian University, 81377 Munich, Germany; 4Division of Gastroenterology and Hepatology, Department of Medicine III, Medical University of Vienna, 1090 Vienna, Austria; 5Liver Cancer (HCC) Study Group Vienna, Division of Gastroenterology and Hepatology, Department of Medicine III, Medical University of Vienna, 1090 Vienna, Austria; 6Department of Gastroenterology, Hepatology and Infectious Diseases, Otto-Von Guericke University Hospital, 39120 Magdeburg, Germany; 7Center of Hematology and Oncology Bethanien, 60389 Frankfurt, Germany; 8University Cancer Center Frankfurt, University Hospital Frankfurt, 60590 Frankfurt, Germany; 9Frankfurt Cancer Institute, Goethe University Frankfurt/Main, 60438 Frankfurt, Germany

**Keywords:** hepatocellular carcinoma, liver transplantation, tyrosine kinase inhibitor, immunosuppression

## Abstract

**Simple Summary:**

Liver transplant is an approach for the curative treatment of early stage hepatocellular carcinoma and the often underlying liver cirrhosis. However, a recurrence of the cancer is to be expected in 10 to 20% of cases after liver transplantation. The treatment of recurrent hepatocellular carcinoma remains difficult, as immunotherapy is usually not recommended for systemic treatment after transplantation. Therefore, it is often necessary to resort to tyrosine kinase inhibitors in advanced stage hepatocellular carcinoma. This multicenter trial aims at assessing the effect of tyrosine kinase inhibitors and immunosuppression on the outcome of patients with recurrent hepatocellular carcinoma after liver transplantation.

**Abstract:**

Introduction: The tyrosine kinase inhibitors (TKIs) sorafenib and lenvatinib represent the first-line systemic therapy of choice for patients with hepatocellular carcinoma (HCC) recurrence after liver transplantation (LT). Under sorafenib and lenvatinib, HCC patients have shown increasingly improved overall survival in clinical studies over the years. In contrast, data on overall survival for patients with HCC recurrence after LT under TKIs are scarce and limited to small retrospective series. In this retrospective, multicenter study, we investigated the efficacy of TKI therapy and the influence of immunosuppression in patients with HCC recurrence after LT. Methods: Retrospective data were collected from four transplant centers from Germany and Austria. We included patients with HCC recurrence after LT between 2007 and 2020 who were treated with a TKI. Results: In total, we analyzed data from 46 patients with HCC recurrence after LT. The most common underlying liver disease was hepatitis C, accounting for 52.2%. The median time to relapse was 11.8 months (range 0–117.7 months). The liver graft was affected in 21 patients (45.7%), and 36 patients (78.3%) had extrahepatic metastases at initial diagnosis of recurrence, with the lung being the most commonly affected (*n* = 25, 54.3%). Of the total, 54.3% (*n* = 25) of the patients were initially treated locally; 39 (85.8%) and 7 (15.2%) patients received sorafenib and lenvatinib, respectively, as first-line systemic therapy. Median overall survival of the whole cohort was 10.9 months (95% confidence interval (95% CI) 6.9–14.9 months) and median progression free survival was 5.7 months (95% CI 2.0–9.4 months) from treatment initiation. Conclusion: Since history of liver transplantation is considered a contraindication for immunotherapy, prognosis of patients with HCC recurrence after LT remains poor.

## 1. Introduction

Liver transplantation (LT) is the only option to cure hepatocellular carcinoma (HCC) and to simultaneously eradicate the usually underlying liver disease. Although the organ allocation criteria also consider factors that indicate a high risk for HCC recurrence, such as macroscopic vascular invasion, HCC recurrence after LT still occurs in 10 to 20% of cases [1].

In recent years, several new therapeutic options have emerged for the treatment of irresectable HCC. The antiproliferative and antiangiogenic tyrosine kinase inhibitor (TKI) sorafenib has been approved for advanced HCC since 2008. In the SHARP trial, the agent showed a significant prolongation of overall survival (OS) by 2.8 months (10.7 months with sorafenib vs. 7.9 months with placebo; hazard ratio (HR) 0.69) [2].

After being the only available agent for a decade, in 2018, lenvatinib proved for the first time to be non-inferior to sorafenib in first-line therapy with respect to the primary endpoint of median overall survival (mOS) (13.6 months (95% CI 12.1–14.9) vs. 12.3 months (95% CI 10.4–13.9); HR 0.92) [3].

Since 2020, the option of immune checkpoint blockade has been available in therapy of HCC. The combination of the anti-programmed cell death ligand 1 (PD-L1) antibody atezolizumab and the anti-VEGF antibody bevacizumab is the first therapy regimen to show prolonged overall survival compared to sorafenib (19.2 months (95% CI 17.0–23.7) with atezolizumab plus bevacizumab and 13.4 months (95% CI 11.4–16.9) with sorafenib (HR 0.66; *p* < 0.001)), as well as prolonged progression-free survival (PFS, 6.9 months (95% CI 5.7–8.6) vs. 4.3 months (95% CI 4.0–5.6; HR 0.65; *p* < 0.001)) [4,5].

In October 2022, the Food and Drug Administration (FDA) has granted approval for the combination of the anti-PD-L1 antibody durvalumab with a single “priming” dose of the anti-cytotoxic T-lymphocyte-associated protein 4 (CTLA-4) antibody tremelimumab (STRIDE regimen) for first-line treatment of advanced HCC due to the positive results in the phase 3 trial [6]. The European Medicines Agency (EMA) granted approval only few months later in February 2023. Regarding median overall survival, durvalumab and tremelimumab (D + T) showed superiority to sorafenib (16.43 months (95% CI 14.16–19.58) versus 13.77 months (95% CI 12.25–16.13; HR 0.78; *p* = 0.0035)).

The recently published first results from CheckMate 9DW revealed significantly prolonged mOS under nivolumab plus ipilimumab compared to lenvatinib or sorafenib [7], so that further options in immuno-oncology therapy of HCC can be expected for the future.

HCC recurrence can be considered a systemic disease due to persistence of malignant cells after resection of the affected liver. Therefore, HCC recurrence after LT often requires systemic therapy [8]. 

Although immune checkpoint inhibitors (ICI) have gained immense importance in the therapy of HCC, they should only be used in exceptional cases for recurrent HCC after LT [9]. Therefore, TKIs usually remain the drug of choice in systemic first-line therapy [10,11,12].

In this multicenter study, we investigated the impact of systemic tumor therapy and immunosuppression on the outcome of patients with HCC recurrence after LT.

## 2. Methods

### 2.1. Patients

We included patients from four tumor centers in Germany and Austria whose HCC recurrence was treated with tyrosine kinase inhibitors after LT between February 2007 and September 2020 (Figure 1). Organ allocation and transplantation followed the standard procedures of organ transplantation in Europe, led by Euro transplant. Listing was carried out accordingly to the commonly accepted clinical and biochemical criteria for LT, including the Milan criteria for HCC and MELD score to assess liver function.

### 2.2. Treatment and Outcome Measurement

Sorafenib and lenvatinib were administered at the discretion of the attending physicians in consideration of the standard dose described in the SHARP trial (sorafenib 800 mg/d) [2] and in the REFLECT trial [3] (lenvatinib 12 mg/day (patients with less than 60 kg body weight 8 mg/day)). 

Radiological response was recorded by computed tomography (CT) or magnetic resonance imaging (MRI). Tumor response was assessed according to the Response Evaluation Criteria in Solid Tumors (RECIST) V1.1 [13] or modified RECIST (mRECIST) [14].

### 2.3. Study Design and Statistical Analysis

Data on the medical history and subsequent course of the disease, including laboratory findings and radiological follow-up, were retrospectively collected from the patient records.

Data on baseline characteristics and radiological response were summarized using descriptive statistics. Continuous variables are reported as median and range, and categorical variables are reported as frequencies and percentages. Median duration of therapy was defined as the time from the first to the last administration of medication.

All patients were followed up until death or last contact. The primary endpoint was overall survival (OS), and secondary endpoints were progression-free survival (PFS) and response rates. Overall survival (OS) was defined as the time from the date of first therapy administration to death. Progression-free survival (PFS) was defined as the time from the date of first therapy administration to radiologic progression or death, whichever occurred first.

Patients who were alive at the time of last contact or data cut-off and did not have radiologically confirmed progression were censored. Survival curves were calculated using the Kaplan–Meier method and compared using the log-rank test (Mantel–Haenszel version). Binary logistic regression models were used to investigate the relationship between independent variables and a binary dependent variable. To analyze prognostic parameters, Cox regression models with forward stepwise likelihood ratio were used. Statistical analyses were performed using R (version 4.2.1) and SPSS (IBM SPSS Statistics 27). *p* values < 0.05 were considered significant. 

This study was performed in accordance with the 1975 Declaration of Helsinki. The retrospective analysis was approved by the local Ethics Committee (SGI03/18, Amendment 01/19) as well as the Ethics Committees of the individual centers.

## 3. Results

### 3.1. Baseline Data

In total, we analyzed data from 46 patients. Of these, 36 (78.3%) were male. The median age at diagnosis of recurrence was 57.5 years (range 44–73 years). The most common underlying liver disease was hepatitis C, accounting for 52.2% of the patients.

Microvascular invasion was evident in 18 patients at the time of transplantation. Eighteen patients (39.1%) had tumor stage T2; fifteen patients (32.6%) had stage T3. Eleven patients (23.9%) had tumor manifestations within the Milan criteria [15]; eighteen (39.1%) were within the up-to-seven criteria [16] at the time of LT.

Before relapse occurred, 38 patients (82.6%) received an immunosuppression regimen based on calcineurin inhibitors (tacrolimus *n* = 36 (78.3%), ciclosporin *n* = 2 (4.3%)). Thirteen patients (28.3%) were immunosuppressed with a mammalian target of rapamycin (mTOR) inhibitor (everolimus *n* = 9 (19.6%), sirolimus *n* = 5 (10.9%)). Six patients (13.0%) had combined a calcineurin inhibitor (CNI) and an mTOR inhibitor.

Fourteen patients (30.4%) received monotherapy with a calcineurin inhibitor, and eighteen (39.1%) had combination therapy with calcineurin inhibitor and MMF (*n* = 16) and/or glucocorticoid (*n* = 7). Five patients (10.9%) were immunosuppressed with an mTOR inhibitor only, and two patients (4.3%) received a combination of mTOR inhibitors and MMF (*n* = 1) and/or glucocorticoid (*n* = 2). In one patient, baseline immunosuppression was unknown (2.2%) (Table 1).

### 3.2. HCC Recurrence after Liver Transplantation

The median time to relapse was 11.8 months (range 0–117.7 months). The liver graft was affected in 21 patients (45.7%), and 36 patients (78.3%) had extrahepatic metastases at initial diagnosis of recurrence, with the lung being the most commonly affected (*n* = 25, 54.3%). Eleven of these patients (23.9%) had both hepatic and extrahepatic manifestations of tumor recurrence. Accordingly, BCLC (Barcelona Clinic Liver Classification) [17] C was predominant with 82.6% (*n* = 38). The binary logistic regression model did not reveal any significant correlation between extrahepatic tumor manifestation and the influencing variables “outside Milan” and “outside up-to-seven” (*p* = 0.144; Nagelkerkes R^2^ = 0.126). 54.3% (*n* = 25) of tumor recurrences were initially treated locally with resection performed in 15 patients (23.6%) and TACE in 5 patients (10.8%). Systemic therapy was initiated with sorafenib in 84.8% (*n* = 39), and the other 15.2% (*n* = 7) received first-line systemic therapy with lenvatinib. Nine patients (19.6%) received the full dose of 800 mg/day sorafenib or 12 mg/day (patients with less than 60 kg body weight 8 mg/day) lenvatinib. The others were given a reduced dose.

When systemic first-line therapy was initiated, most of the patients showed acceptable liver function (Child Pugh A in 42 patients (91.3%)).

At the start of systemic first-line treatment, 10 patients (21.7%) received monotherapy with a calcineurin inhibitor, and 6 (13.0%) had combination therapy with a calcineurin inhibitor and MMF (*n* = 4) and/or glucocorticoid (*n* = 3). Six patients (13.0%) were immunosuppressed with an mTOR inhibitor only, and four (8.7%) patients received a combination of mTOR inhibitors and MMF (*n* = 3) and/or glucocorticoid (*n* = 3). Twenty patients (43.5%) had combined a calcineurin inhibitor and a mTOR inhibitor (Table 2).

### 3.3. Survival Data under Tyrosine Kinase Inhibitors Sorafenib or Lenvatinib

Altogether, patients showed a median overall survival (mOS) of 10.9 months (95% confidence interval (95% CI) 6.9–14.9 months) and a median progression-free survival (mPFS) of 5.7 months (95% CI 2.0–9.4 months) from treatment initiation (Figure 2). Median time to progression (TTP) was 3.5 months (range 1.1–38 months). Treatment with TKIs achieved a disease control rate (DCR) of 17.4%. Twenty-three patients (50%) showed progressive disease as best response on TKIs. In 15 patients (32.6%), response could not be assessed, in most cases due to early discontinuation of treatment as a result of adverse events or death of the patient (Table 3).

### 3.4. Survival Data in Dependence on the Immunosuppression

Survival data of patients taking an mTOR inhibitor-based immunosuppression regimen were compared with those not taking a mTOR inhibitor. 

We compared the overall survival of patients who were already immunosuppressed by a mTOR inhibitor before the recurrence of HCC was diagnosed with those patients whose immunosuppression did not contain a mTOR inhibitor before detection of recurrence. mOS from the detection of recurrence was 22.2 months (95% CI 0.0–49.7 months) in the group of patients with mTOR inhibitor. Patients in the cohort without mTOR inhibitor showed a mOS of 12.7 month (95% CI 8.8–16.6 months, *p* = 0.376) (Figure 3).

As Figure 4 shows, in many patients, a switch to mTOR inhibitor-based immunosuppression was performed after the detection of HCC recurrence.

### 3.5. Survival Data in Dependence on the Time of Recurrence and the Organ Manifestation

Likewise, survival data showed no significant difference when comparing patients having extrahepatic tumor manifestations with patients having intrahepatic manifestation of HCC recurrence only (14.4 months (95% CI 10.5–18.4 months) vs. mOS 12.7 months (95% CI 0.0–36.9 months); HR 0.985; *p* = 0.966). 

Similarly, there was no significant difference in survival between patients with early HCC recurrence (≤24 months after LT, *n* = 33, median time to recurrence: 8.3 months) and patients with late HCC recurrence (>24 months after LT, *n* = 13, median time to recurrence: 36.8 months), showing a median survival of 12.6 months (95% CI 10.3–14.9 months) vs. 24.9 months (95% CI 6.3–43.5 months; HR 0.616 (0.313–1.209); *p* = 0.155).

Ten patients received second-line systemic therapy, including four regorafenib, three ramucirumab, and three cabozantinib.

## 4. Discussion

Patients with HCC recurrence after LT represent a special subgroup among patients with HCC. Whereas the usually underlying liver cirrhosis is a crucial prognostic factor, patients after LT usually have significantly improved liver function.

On the other hand, the patients are now immunosuppressed, which can cause an increased growth tendency of the tumor [18] and limits the use of immuno-oncology (IO) therapy, as simultaneous use of immunosuppressants reduces the effect of IO therapy. In addition, there is an increased risk of rejection due to the mode of action, which is based on amplification of an immune response [19]. In particular, the PD1/PD-L1 pathway, which is inhibited by atezolizumab and durvalumab, contributes to the development of allograft tolerance [20]. 

However, this transplant tolerance maintained by immunosuppressant use also results in an increased risk of HCC recurrence and increased tumor growth tendency. It is possible that the various immunosuppressants differ in terms of their pro-oncologic properties [21,22]. mTOR inhibitors have often been attributed an antineoplastic effect, as the PI3K/Akt/mTOR pathway is considered an important regulator of cell proliferation and angiogenesis and has been shown to be a factor in hepatocarcinogenesis [23]. However, data demonstrating a sufficient effect of mTOR inhibitors on overall and progression-free survival of patients with HCC recurrence after LT are currently lacking [24,25].

Therefore, even after the success of atezolizumab + bevacizumab and durvalumab + tremelimumab in a phase 3 trial, TKIs remain the standard of care in systemic therapy of recurrent HCC after LT.

In this retrospective, multicenter study, we investigated the impact of immunosuppression and systemic tumor therapy with TKIs on the outcome of patients with HCC recurrence after LT. 

Median overall survival (mOS) under TKI treatment (84.8% sorafenib, 15.2% lenvatinib) was comparable with mOS under sorafenib in the SHARP trial [2]. This may be related to the fact that although liver cirrhosis was eradicated as a prognosis-limiting underlying disease in most cases, the patients had a suppressed immune system instead.

Tovoli et al. analyzed data of 81 patients who were treated with sorafenib for HCC recurrence after transplantation from 2010 onwards [26]. They reported a median overall survival of 19 months, which is substantially longer than in our cohort. One reason for this could be that a higher proportion received second-line therapy in the multicenter Italian study. It should be noted here that second- and third-line therapies have become increasingly available over the past few years. Overall, the published survival time data under sorafenib after transplantation are very heterogeneous: while mOS of up to 20 months is also reported in further cases [27,28], considerably shorter mOS is also documented by other authors [11,29].

At a DCR of 17.4%, patients in our study showed a response to therapy with TKIs that was far behind the response to therapy with sorafenib (DCR 43% (SHARP); 60.5% (REFLECT)) and lenvatinib (DCR 75.5% (REFLECT)) in phase 3 trials [2,3]. This may be due to an increased tumor growth tendency in the liver transplanted and thus immunosuppressed patients compared to patients without LT in the phase 3 study. Another explanation could be the low rate of fully dosed TKI. Only less than 20% of the included patients received the in the REFLECT trial established dose of 800 mg/day sorafenib or 12 mg/day (patients with less than 60 kg body weight 8 mg/day) lenvatinib. An Asian–Italian study evaluating the efficacy and safety of lenvatinib in patients with recurrent HCC after LT showed an excellent response rate to lenvatinib (DCR 88.9%) [10]. However, considerably more patients received the standard dose of lenvatinib compared to our trial.

A further reason for the poor response to therapy with TKIs in our study may be the time of therapy initiation. TKIs had shown increasingly better survival data and response rates in the various phase 2 and 3 trials over the last years [2,3,4]. Patients in our study were treated with TKIs between 2007 and 2021, whereas the treatment period in the Asian–Italian study was much later between 2017 and 2021.

Kornberg et al. [30] detected late HCC recurrence (>24 months) as an independent predictor of survival after HCC relapse. Our data indicate a trend toward longer survival at late recurrence (>24 months), but without statistical significance.

Although survival analysis showed no significant differences between patients taking a mTOR inhibitor-based immunosuppression regimen and those not taking a mTOR inhibitor, it is possible that the Kaplan–Meier curves indicate a slight advantage for the immunosuppression based on mTOR inhibitors in terms of overall survival. It should be noted that due to the small sample size and the skewed distribution across both groups, there is a higher probability that an existing effect cannot be detected. Prospective trials with higher numbers of cases are required.

Sirolimus and everolimus are currently available as mTOR inhibitors for immunosuppression after LT. Everolimus is a derivative of sirolimus with improved bioavailability. mTOR inhibitors have long been thought to have antineoplastic properties [23]. In the treatment of HCC, everolimus failed to achieve a satisfactory outcome in a phase 1/2 study, so that this substance was not further investigated in the treatment of HCC [31]. However, patients after solid organ transplantation were excluded and higher antiproliferative doses of mTOR inhibitors were administered. So far, no positive effect on tumor progression and overall survival could be achieved for mTOR inhibitors as immunosuppression in recurrent HCC after liver transplantation. Further investigations are needed here: prospective studies could probably help to better assess the benefits mTOR inhibitors in the prevention of HCC recurrence after LT, but also in the treatment of recurrent HCC after transplant [32].

## 5. Conclusions

The prognosis for non-resectable recurrent HCC after liver transplantation remains poor. In our study, we could not demonstrate any significant benefit of mTOR inhibitors with regard to overall survival. However, prospective studies with larger sample sizes are still required. TKIs are usually administered as first-line systemic treatment. However, data on further treatment options as alternative to TKIs or second-line therapy after TKIs is scarce.

## Figures and Tables

**Figure 1 cancers-16-02442-f001:**
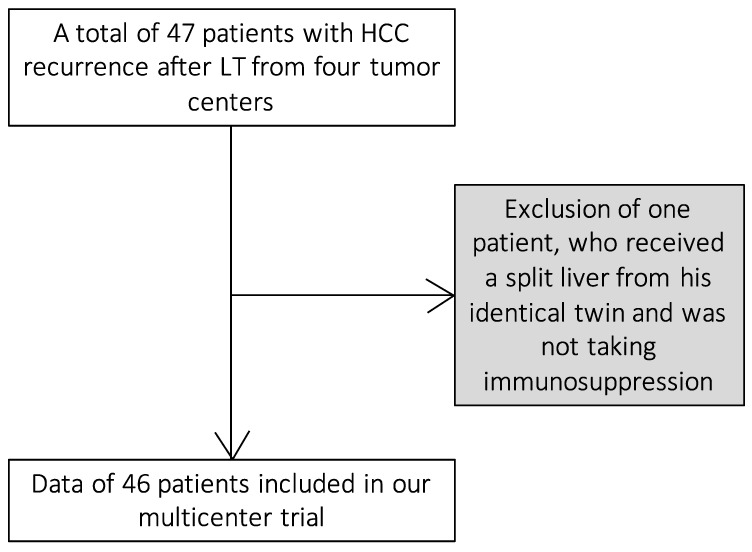
Inclusion of the patients in our study.

**Figure 2 cancers-16-02442-f002:**
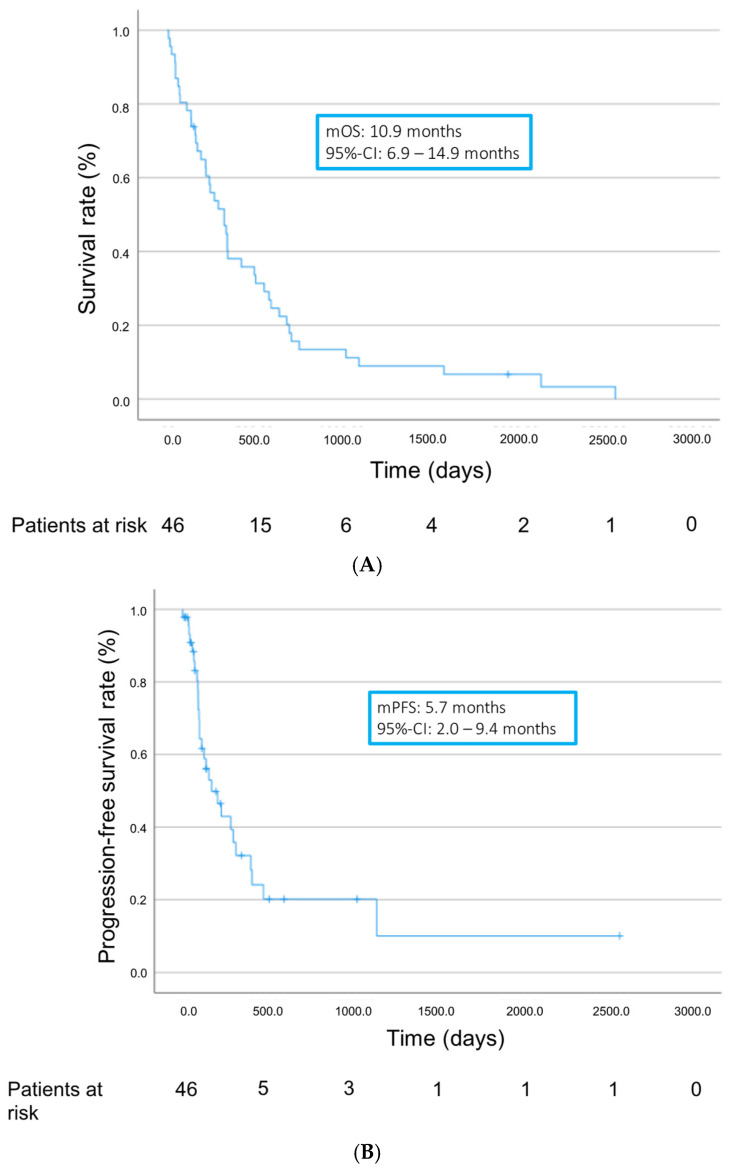
Overall survival (**A**) and progression-free survival (**B**) from start of TKI treatment.

**Figure 3 cancers-16-02442-f003:**
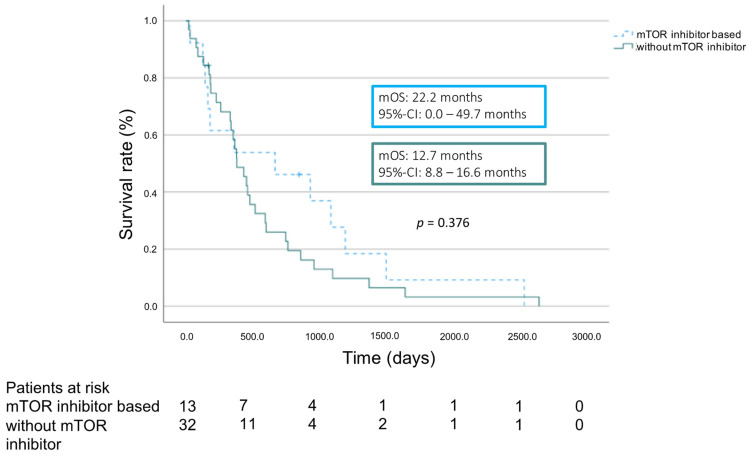
Overall survival according to immunosuppression regimen.

**Figure 4 cancers-16-02442-f004:**
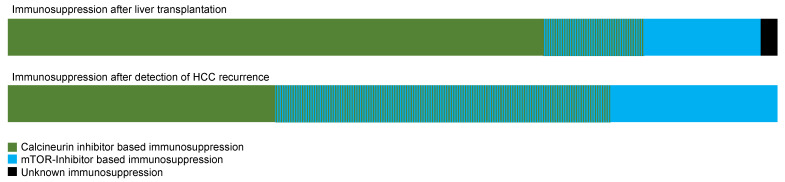
Immunosuppression before and after detection of HCC recurrence.

**Table 1 cancers-16-02442-t001:** Patients’ baseline data.

Demographics	
Patients; *n*	46
Male/female; *n*, (%)	36 (78.3)/10 (21.7)
Age at recurrence [y]; median, (range)	57.5 (44–73)
BMI; median, (range)	25.4 (18.1–41.5)
Etiology of liver disease	
Alcohol; *n*, (%)	10 (21.7)
NASH/NAFLD; *n*, (%)	6 (13.0)
Hepatitis B; *n*, (%)	8 (17.4)
Hepatitis C; *n*, (%)	24 (52.2)
Autoimmune hepatitis; *n*, (%)	1 (2.2)
Prior therapy before oLT; *n*, (%)	35 (76.1)
Resection; *n*, (%)	4 (8.7)
Ablation; *n*, (%)	7 (15.2)
TACE; *n*, (%)	31 (71.7)
Tumor stage at oLT	
T1; *n*, (%)	8 (17.4)
T2; *n*, (%)	18 (39.1)
T3; *n*, (%)	15 (32.6)
T4; *n*, (%)	2 (4.3)
Unknown; *n*, (%)	3 (6.5)
Tumor grade at oLT	
1; *n*, (%)	2 (4.3)
2; *n*, (%)	20 (43.5)
3; *n*, (%)	12 (26.1)
Unknown; *n*, (%)	12 (26.1)
Within the Milan criteria; *n*, (%)	11 (23.9)
Within the up-to-seven criteria; *n*, (%)	18 (39.1)
AFP at oLT [ng/mL]; median, (range)	59.2 (1.0–60,500.0)
Immunosuppression before HCC recurrence	
Ciclosporin; *n*, (%)	2 (4.3)
dose [mg/d]; median, (range)	140 (80–200)
Tacrolimus; *n*, (%)	36 (78.3)
dose [mg/d]; median, (range)	4 (0.5–10)
Sirolimus; *n*, (%)	5 (10.9)
dose [mg/d]; median, (range)	2 (2–3)
Everolimus; *n*, (%)	9 (19.6)
dose [mg/d]; median, (range)	2,5 (1–6)
Mycophenolat mofetil; *n*, (%)	17 (37.0)
dose [mg/d]; median, (range)	1000 (500–2000)
Dexamethason; *n*, (%)	1 (2.2)
dose [mg/d]	8
Prednisolon; *n*, (%)	8 (17.4)
dose [mg/d]; median, (range)	6.25 (2.5–20)
Unknown; *n*, (%)	1 (2.2)

BMI: body mass index; NASH: non-alcoholic steatohepatitis; NAFLD: non-alcoholic fatty liver disease; oLT: orthotopic liver transplantation; TACE: transarterial chemoembolization; AFP: alpha-fetoprotein.

**Table 2 cancers-16-02442-t002:** Management of HCC recurrence.

Time to relapse [months]; median, (range)	11.8 (0–117.7)
Organs affected by recurrence	
Liver graft; *n*, (%)	21 (45.7)
Extrahepatic metastases; *n*, (%)	36 (78.3)
Lung; *n*, (%)	25 (54.3)
Bone; *n*, (%)	7 (15.2)
Peritoneum; *n*, (%)	7 (15.2)
Adrenal gland; *n*, (%)	5 (10.9)
Intra- and extrahep. recurrence; *n*, (%)	11 (23.9)
BCLC stage	
A; *n*, (%)	3 (6.5)
B; *n*, (%)	5 (10.9)
C; *n*, (%)	38 (82.6)
D; *n*, (%)	0 (0)
Local treatment; *n*, (%)	25 (54.3)
Resection; *n*, (%)	15 (32.6)
Ablation; *n*, (%)	2 (4.34)
TACE; *n*, (%)	5 (10.8)
SIRT; *n*, (%)	4 (8.7)
Systemic first-line therapy	
Sorafenib; *n*, (%)	39 (84.8)
Initial dose [mg/d]; median, (range)	400 (200–800)
Lenvatinib; *n*, (%)	7 (15.2)
Initial dose [mg/d]; median, (range)	4 (4–12)
Immunosuppression under first-line treatment	
Ciclosporin; *n*, (%)	2 (4.3)
dose [mg/d]	200
Tacrolimus; *n*, (%)	34 (73.9)
dose [mg/d]; median, (range)	3 (0.5–10)
Sirolimus; *n*, (%)	11 (23.9)
dose [mg/d]; median, (range)	1 (0.5–2)
Everolimus; *n*, (%)	19 (41.3)
dose [mg/d]; median, (range)	2 (0.5–6)
Mycophenolat mofetil; *n*, (%)	7 (15.2)
dose [mg/d]; median, (range)	1000 (500–2000)
Dexamethason; *n*, (%)	1 (2.2)
dose [mg/d]	8.0
Prednisolon; *n*, (%)	5 (10.9)
dose [mg/d]; median, (range)	5.0 (2.5–10)
Laboratory results at treatment initiation	
AFP [ng/mL]; median, (range)	59.2 (1.0–60,500)

BCLC: Barcelona Clinic Liver Classification; TACE: transarterial chemoembolization; SIRT: selective internal radiation therapy; AFP: alpha-fetoprotein.

**Table 3 cancers-16-02442-t003:** Radiological response and survival data during first-line treatment.

Response to TKI Therapy, *n* = 46	
mOS [months]; (95% CI)	10.9 (6.9–14.9)
mPFS [months]; (95% CI)	5.7 (2.0–9.4)
Stable disease; *n*, (%)	8 (17.4)
Progressive disease; *n*, (%)	23 (50.0)
No staging performed; *n*, (%)	15 (32.6)
Disease control rate [%]	17.4
mTTP [months]; median, (range)	3.5 (1.1–38)

## Data Availability

Data are contained within the article.

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
