# Peer review of "Systemic Treatment of Recurrent Hepatocellular Carcinoma after Liver Transplantation: A Multicenter Trial"

_cancers, 2024, doi:10.3390/cancers16132442_

Round 1

Reviewer 1 Report

Comments and Suggestions for Authors

The authors present a study to evaluate the impact of immunosuppression and systemic treatment of the tumor on the outcome of patients with HCC recurrence after LT

 The article is well-written, but there are certain modifications that could enhance comprehension.

The introduction section should be expanded with new references.

In the methods section, a diagram could be added to more clearly explain the objective of the study.

It would be advisable to add a table in the results section with all the statistical analysis, to properly visualize the weight of the variables in the study. Additionally, it is necessary to better explain this section for greater reader comprehension.

A conclusion at the end of the study, summarizing its contributions, is missing.

Author Response

The introduction section should be expanded with new references.

Thank you for your recommendations. According to this recommendation we have added two references relating to therapy of recurrent HCC with TKI. We have also added a paragraph on the latest developments in HCC therapy.

In the methods section, a diagram could be added to more clearly explain the objective of the study.

We added a graph (Figure 1) to visualize the characteristics of the included patients.

It would be advisable to add a table in the results section with all the statistical analysis, to properly visualize the weight of the variables in the study. Additionally, it is necessary to better explain this section for greater reader comprehension.

We are not exactly sure what the reviewer suggests here. Only basic statistical testing was used in this manuscript as in a lot of manuscripts before from our group. As there is no unique special approach in this work we think the statistical analysis section explains everything, however we tried to be more clear here in the reviewed part.

A conclusion at the end of the study, summarizing its contributions, is missing.

A conclusion at the end of the manuscript has been added.

Reviewer 2 Report

Comments and Suggestions for Authors

Comments:

The authors of this study investigated the efficacy of TKI therapy in patients with HCC recurrence after LT. Retrospective data were collected from four transplant centers from Germany and Austria. Their results showwed that the median overall survival of the whole cohort was 10.9 months and median progression free survival was 5.7 months from treatment initiation and indicated that the prognosis of patients with HCC recurrence after LT remains poor. The subject of this manuscript is of value, but there are a few of defects need to be modified.

1. When the term first appears, the full name and abbreviation need to be provided, and then the abbreviation should be used. Please check:liver transplantation and LT in the abstract and full text sections. Many “liver transplantation should be replaced by the abbreviation “LT”. overall survival (OS) and progression free survival (PFS) should also be checked and revised.

2. Abstract section: 54.3% (n=25) of tumor recurrences were initially treated locally. This sentence may be misleading, here:54.3% (n=25) of tumor recurrences: referring 25 patients or 25 tumors. Are there any patients with multiple nodules.

3. Introduction section: In this multicenter study, we investigated the impact of immunosuppression and systemic tumor therapy on the outcome of patients with HCC recurrence after LT. I suggest that the order of “immunosuppression and systemic tumor therapy” be consistent throughout the entire text. Based on the content of the title, it is recommended to first describe “systemic tumor therapy”.

4. Methods section:Radiological response was recorded by computed tomography (CT) or magnetic resonance imaging (MRI). Tumor response was assessed according to the Response Evaluation Criteria in Solid Tumors (RECIST) V1.1 or modified RECIST (mRECIST). Why did the author use two Criteria to evaluate treatment response? Whether the mRECIST be uniformly used. The author should explain clearly.

5. Table 2. Management of HCC recurrence.The total number of cases receiving various local treatment methods (Resection,Ablation,TACE,SIRT)(26) is not equal to the total number (described 25). Please check and revise. Very important.

Author Response

  1. When the term first appears, the full name and abbreviation need to be provided, and then the abbreviation should be used. Please check: liver transplantation and LT in the abstract and full text sections. Many “liver transplantation” should be replaced by the abbreviation “LT”. “overall survival (OS) and progression free survival (PFS)” should also be checked and revised.

Thank you for pointing this out. We agree with this comment, therefore we replaced further “liver transplantation” by “LT”. We also revised “overall survival” (OS) and “progression free survival” (PFS).

  1. Abstract section: 54.3% (n=25) of tumor recurrences were initially treated locally. This sentence may be misleading, here:54.3% (n=25) of tumor recurrences: referring 25 patients or 25 tumors. Are there any patients with multiple nodules?

There are some patients with multiple nodules. This sentence refers to 25 patients. We revised this sentence according to the recommendation (page 1, line 41).

  1. Introduction section: In this multicenter study, we investigated the impact of immunosuppression and systemic tumor therapy on the outcome of patients with HCC recurrence after LT. I suggest that the order of “immunosuppression” and “systemic tumor therapy” be consistent throughout the entire text. Based on the content of the title, it is recommended to first describe “systemic tumor therapy”.

We changed this sentence into “In this multicenter study, we investigated the impact of systemic tumor therapy and immunosuppression on the outcome of patients with HCC recurrence after LT.”(page 2, line 93/94).

  1. Methods section: Radiological response was recorded by computed tomography (CT) or magnetic resonance imaging (MRI). Tumor response was assessed according to the Response Evaluation Criteria in Solid Tumors (RECIST) V1.1 or modified RECIST (mRECIST). Why did the author use two Criteria to evaluate treatment response? Whether the mRECIST be uniformly used. The author should explain clearly.

In > 95% mRECIST was used. However, as it was a long retrospective study from several centers RECIST was used as well as in most HCC publications from the last years. However, as long as one center sticks to one method outcome parameters will not change.

  1. Table 2. Management of HCC recurrence. The total number of cases receiving various local treatment methods (Resection, Ablation, TACE, SIRT)(26) is not equal to the total number (described 25). Please check and revise. Very important.

The discrepancy between the total number of patients who received locoregional therapy (n=25) and the number of interventions performed can be explained by the fact that in some patients several methods were performed.